# Biomarker-Based Preoperative Risk Stratification for Patients Undergoing Non-Cardiac Surgery

**DOI:** 10.3390/jcm9020351

**Published:** 2020-01-27

**Authors:** Timur Yurttas, Reka Hidvegi, Miodrag Filipovic

**Affiliations:** Division of Anesthesiology, Intensive Care, Rescue and Pain Medicine, Kantonsspital St.Gallen, 9001 St. Gallen, Switzerland; timur.yurttas@kssg.ch (T.Y.); reka.hidvegi@kssg.ch (R.H.)

**Keywords:** preoperative risk stratification, cardiac biomarkers, perioperative myocardial infarction/injury

## Abstract

Perioperative morbidity and mortality remains a substantial problem and is strongly associated with patients’ cardiac comorbidities. Guidelines for the cardiovascular assessment and management of patients at risk of cardiac issues while undergoing non-cardiac surgery are traditionally based on the exclusion of active or unstable cardiac conditions, determination of the risk of surgery, the functional capacity of the patient, and the presence of cardiac risk factors. In the last two decades, strong evidence showed an association between cardiac biomarkers and adverse cardiac events, with newer guidelines incorporating this knowledge. This review describes a biomarker-based risk-stratification pathway and discusses potential treatment strategies for patients suffering from postoperative myocardial injury or infarction.

## 1. Background

Postoperative deaths are one of the most important causes of death worldwide [1]. In Europe, perioperative mortality is high (3% after elective, 5% after urgent, and 10% after emergency surgery) [2]. The most important comorbidities associated with postoperative death are cirrhosis, congestive heart failure and coronary artery disease [2]. In a large database study of more than 600,000 patients, those with heart failure undergoing complex surgery had a 90-day mortality of more than 10% [3]. Based on the strong association of cardiac diseases with perioperative mortality, many efforts were made to stratify the risk and optimize the outcomes of patients at risk of cardiac issues when undergoing major non-cardiac surgery. The American and European cardiological and anesthesiological societies’ guidelines are currently based on four steps:Exclusion of active or unstable cardiac conditions;Determination of risk and complexity of the surgical procedure;Determination of the patient’s functional capacity (physical fitness);Determination of cardiac risk factors.

If increased cardiac risk is determined, patients are assigned for a cardiological work-up, including testing for myocardial ischemia or important cardiac valve diseases. However, evidence demonstrating a clear benefit of this complicated and time- and resource-consuming process is scarce. Accordingly, new approaches for cardiac risk stratification are under consideration. The Canadian Cardiovascular Society proposed a biomarker-based algorithm, putting the preoperative measurement of brain natriuretic peptide (BNP or NT-proBNP) at the center of risk stratification [4]. In case of increased natriuretic peptide, the guidelines call for postoperative troponin surveillance. This review discusses the traditional and biomarker-based approaches for risk stratification and proposes a synthesis of both. In addition, we propose a strategy to treat perioperative myocardial injury as detected by serial troponin measurements.

## 2. Traditional Risk Stratification

### 2.1. Exclusion of Active or Unstable Cardiac Conditions

Active or unstable cardiac conditions include ongoing myocardial ischemia (i.e., unstable angina or current or very recent myocardial infarction), decompensated heart failure, severe arrhythmia (e.g., atrial fibrillation with uncontrolled ventricular rate, high grade atrioventricular block) and symptomatic valvular heart disease (e.g., symptomatic aortic stenosis) [5]. In these situations, patients should be referred for an immediate cardiological work-up and restabilization. The treatment strategies correspond to the current cardiological guidelines and consequently can cause substantial delay in scheduled surgery. Restabilization of decompensated heart failure can last for several months [5]. Importantly, indications for coronary revascularization also follow the usual cardiological guidelines. More details can be found in [5]. 

### 2.2. Determination of the Risk of the Scheduled Surgical Procedure

Based on the risk for perioperative adverse cardiac events, non-cardiac surgical procedures are grouped into three classes: Low-risk procedures (cardiac risk of <1%) include minor surgery on the body surface, high-risk procedures (>5%) include major thoracic, abdominal, and vascular surgery, and the intermediate group (cardiac risk of 1–5%) includes minor intraperitoneal, intrathoracic, and vascular procedures, surgery of the head and neck, and major orthopedic and neurosurgical operations [5]. Patients free from active or unstable cardiac conditions scheduled for low-risk surgeries can undergo the procedure without further evaluation. In contrast, those scheduled for intermediate or high-risk procedures require their functional capacity to be considered. 

### 2.3. Determination of Functional Capacity

Functional capacity is expressed as the metabolic equivalent (MET), which is defined as a multiple of basal oxygen consumption (about 3.5 mL O_2_/kg min). Functional capacity can be objectively measured via exercise testing, or estimated from the patient’s ability to perform daily living activities.

The association between low functional capacity and increased morbidity and mortality is robust [6,7], however, evidence for a clear cut-off value is lacking. Therefore, all current guidelines use 4 MET as the cut-off for the definition of low functional capacity. Climbing two flights of stairs or running short distances equals roughly 4 MET. If a patient’s functional capacity is below 4 MET (or if they are not able to exercise because of non-cardiac or non-pulmonary conditions, such as severe osteoarthrosis), further evaluation of the cardiac condition is necessary. In contrast, patients with high functional capacity are at low risk for adverse cardiac events and can undergo surgery. However, further work-up is mandatory in all patients suffering from angina [8,9]. 

The ongoing international multicenter MET REPAIR study is currently recruiting around 15,000 patients and aims to define an evidence-based cut-off value for functional capacity and subsequent adverse cardiac events using a validated questionnaire [10]. 

### 2.4. Determination of Cardiac Risk Factors 

Based on the revised cardiac risk index (RCRI) [11,12], the following comorbidities are associated with adverse perioperative outcomes: Ischemic heart disease (e.g., previous myocardial infarction), any history of heart failure, stroke, or transient ischemic attack, diabetes mellitus requiring insulin treatment, and renal dysfunction (defined as serum creatinine of ≥170 μmol/L or a creatinine clearance of <60 mL/min 1.73 m^2^). If one or several of these conditions are present, further evaluation is recommended. However, in the absence of these, surgery can be performed.

A cardiological work-up depends on assumed cardiac conditions and may include non-invasive testing for myocardial ischemia or echocardiography for presumed valvular diseases. The work-up follows the usual cardiological guidelines. If the tests reveal cardiac conditions that call for invasive treatment, the strategy should be specified using a multi-disciplinary team approach (cardiologist, surgeon, anesthesiologist, intensivist, general practitioner, and so on).

## 3. Biomarker-Based Risk Stratification

Nowadays, cardiac biomarkers are indispensable elements for the diagnosis of acute cardiac conditions, such as acute myocardial infarction or decompensated heart failure. Also, in perioperative medicine, robust evidence proves the importance of biomarkers to predict and detect adverse cardiac events. Currently, the high negative predictive value of non-elevated levels of natriuretic peptides and the very close association between postoperatively elevated troponin levels and morbidity and mortality are of central importance. Newer biomarkers are under investigation [13,14]. However, cardiac biomarkers are of marginal importance in the American [8] and European [9] guidelines for perioperative risk assessment. 

In contrast, the corresponding guidelines of the Canadian Cardiovascular Society place the biomarkers at the center [4]. The Canadian guidelines start with the definition of patients at cardiac risk using broad criteria (≥45 years of age or 18–44 years with known significant cardiovascular disease). Only surgical procedures requiring overnight hospital admission are considered, but the risk is not further classified. The next steps depend on the urgency of surgery. Emergency surgery is performed without any preoperative testing, whereas urgent or semi-urgent call for preoperative cardiac assessment in case of an unstable cardiac condition, suspected undiagnosed severe pulmonary arterial hypertension, or obstructive cardiac disease (e.g., severe aortic stenosis, severe mitral stenosis, or severe hypertrophic obstructive cardiomyopathy). In elective procedures, BNP or NTproBNP are measured if the patent’s age (>65 years) or the RCRI indicates the presence of "significant cardiovascular disease". Postoperatively, for all elderly patients or those with elevated cardiovascular risk, troponin surveillance is recommended daily for 48–72 h. In contrast, troponin is not measured if the preoperative BNP is below 92 ng/mL (or the NTproBNP is below 300 ng/mL) [4,15]. 

A detailed and explanatory graph can be found in the references [16]. 

### 3.1. Brain Natriuretic Peptide

Natriuretic peptides (atrial natriuretic peptide (ANP) and brain natriuretic peptide (BNP)) are released by cardiomyocytes in case of cardiac overload. Natriuretic peptides increase sodium and water excretion and decrease blood pressure. BNP is produced by cleavage of the pre-hormone pro-BNP to the biologically active BNP and the non-biologically active N-terminal (NT)-pro-BNP [17,18]. Laboratory testing is commercially available and well-established for both [19].

In non-surgical patients, BNP is mainly used to diagnose and evaluate heart failure [20]. Preoperatively, the high negative predictive value of non-elevated levels of natriuretic peptides is more important [21]. In an individual patient data meta-analysis of studies using natriuretic peptides, a BNP level of 116 ng/L was detected as optimal for preoperative risk stratification [22]. This cut-off was remarkably close to the value found in a meta-analysis in cardiac patients [23]. The Canadian Society uses cut-off values of 92 ng/L for BNP and 300 ng/L for NT-proBNP, respectively. Patients above these cut-off values undergo troponin surveillance (see below).

Despite the recent meta-analyses regarding the association of natriuretic peptides with perioperative risk, we are not aware of any published study questioning the prognostic significance of this relationship. Very recently, a substudy of the VISION trial [24,25,26], which included more than 10,000 patients, showed a strong association between preoperative NTproBNP levels and postoperative adverse cardiac events [27]. Remarkably, the ongoing PeriOperative ISchemic Evaluation-3 Trial (POISE-3) used a preoperative NTproBNP value of ≥200 ng/L as one of the inclusion criteria indicating cardiac risk [28]. 

### 3.2. Cardiac Troponins

Cardiac troponins are structural proteins of the myocardial contractile apparatus that are released during myocardial injury. First reports on the diagnostic and prognostic importance of troponin release in patients with acute coronary syndromes were published in the 1990s [29], with a vast number of studies confirming the importance of troponin release in many cardiac conditions since then. In addition, advances in laboratory analyses allowed for more specific testing and the development of tight rule-in rule-out protocols for the diagnosis of acute coronary syndromes. The current definition of myocardial infarction is strongly based on elevated cardiac troponins [30]. 

Perioperatively, the association between elevated troponins and adverse post-surgical cardiac outcomes was established in the early years of this millennium [31,32] and was confirmed via meta-analysis [33]. 

The VISION trials performed by Deveraux were landmark studies in perioperative medicine that proved the association between postoperative troponin release and adverse outcomes in a large number of patients [24,25,26]. Equally important, these studies also showed that troponin release was not necessarily accompanied by clinical symptoms (e.g., chest pain) or electrocardiography (ECG) changes. In contrast, ischemic symptoms are found only in around one third of troponin-positive patients [34], meaning that, in a high-risk population, a substantial proportion of troponin-positive patients may stay undetected without systematic troponin surveillance [35]. 

Accordingly, the Canadian guidelines call for troponin surveillance in patients with a cardiac risk profile and preoperative elevated BNP. However, a clear treatment strategy for these patients has not been established so far. In the Basel Perioperative Myocardial Infarction (PMI) study [36], more than 2500 patients were systematically investigated pre- and postoperatively. A postoperative rise in troponin-T was found in 16% of patients, of whom 9% died within 30 days after surgery. In the group of patients free from troponin elevation, the same mortality rate was 2%. In addition to the VISION trials, this study established the importance of preoperative troponin elevation for subsequent adverse cardiac events [36]. Despite being primarily an observational study, the authors followed a clear treatment strategy for patients suffering from perioperative myocardial injury and published a helpful supplemental algorithm. However, this algorithm was not prospectively tested.

The nomenclature for conditions with troponin elevation is somewhat confusing. We recommend the terms used in the “Fourth Universal Definition of Myocardial Infarction” [30]. “Acute Myocardial Injury” is defined as a dynamic change in troponin levels without any signs or symptoms of acute ischemia (e.g., chest pain, dynamic ST-segment changes, new wall motion abnormalities, and so on). In contrast, the presence of ischemic features accompanied by changes in troponin levels defines “Acute Myocardial Infarction”, which is further subdivided in type 1 and type 2. 

Notably, the source of cardiac troponin is always the myocardium, but the cause of troponin release is not always myocardial ischemia. Acute heart failure, pulmonary embolism, sepsis, and thoracic trauma are examples of cardiac and extra-cardiac conditions frequently associated with troponin release [37]. However, the prognoses of patients with “non-ischemic” troponin increases are equally as bad as that of troponin-positive patients of ischemic origin [36].

## 4. The Swiss Algorithm for Perioperative Risk Stratification and Optimization

Based on the previous guidelines, the recent Canadian guidelines, and on the literature described above, a “Swiss Algorithm” was recently published on behalf of the Swiss Society of Anesthesia and Reanimation (SGAR-SSAR) [38]. This algorithm uses a classical 4-step approach (steps highlighted in color in Figure 1: red = step 1, exclusion of active or unstable cardiac conditions; yellow = step 2, determination of risk and complexity of the surgical procedure; green = 3, determination of the patient’s functional capacity (physical fitness); and light pink = 4, determination of cardiac risk factors) but incorporates preoperative BNP (and troponin) measurements and postoperative troponin surveillance (Figure 1). If the risk of surgery, poor functional capacity, and the presence of clinical risk factors indicate higher risk, BNP is used to further differentiate the patient’s risk. Patients with normal levels can undergo surgery without further testing, whereas patients with BNP or NT-proBNP elevation are at high risk and are required to undergo cardiological work-up (in accordance with the current European and American guidelines, but not the new Canadian guidelines). Thereafter, troponin surveillance is recommended. Elevated troponin is defined as any value above the 99th percentile upper reference limit of the test and troponin surveillance is recommended daily for 48 to 72 h. A change of ≥20% is interpreted as significant [39,40]. In case of urgent surgery, biomarker testing is proposed to be performed in a similar way (Figure 2), however, cardiac work-up is not feasible in these cases.

Importantly, the “Swiss Algorithm” has not yet been prospectively tested, but this short-coming is also true for the other guidelines. 

## 5. Treatment Strategies for Myocardial Infarction and Injury

As mentioned, no clear treatment strategy for patients with perioperative troponin release (perioperative myocardial infarction and injury) has been established so far. This is a strong point criticism in every method of troponin surveillance; however, knowledge and experience from cardiological patients suffering from myocardial infarction and pathophysiological facts can help establish a meaningful, yet so far unproven, treatment algorithm.

In case of dynamic troponin changes, a logical first step is to differentiate ischemic from non-ischemic causes of myocardial injury. Clinical examination, ECG, and echocardiography can reveal the presence of features of myocardial ischemia and give evidence for non-ischemic causes of troponin release (e.g., sepsis, pulmonary embolism, trauma, heart failure, myocarditis, and so on). If a non-ischemic origin is likely, the treatment of these patients should focus on the underlying condition.

If myocardial ischemia is present, further differentiation is mandatory. Classical myocardial infarction is caused by plaque rupture and thrombosis (type 1 myocardial infarction). In contrast, an imbalance between myocardial oxygen demand and oxygen supply also causes myocardial ischemia and troponin release (type 2 myocardial infarction). Conditions associated with oxygen supply and demand imbalance, such as tachycardia, hypotension, hypertension, or anemia, are common in the perioperative setting. Accordingly, many patients suffering from perioperative acute myocardial infarction might have a type 2 infarction. From this pathophysiological point of view, it seems reasonable that the treatment strategies for type 1 and type 2 myocardial infarctions are different. In patients with type 1 myocardial infarction, myocardial revascularization must be considered. In contrast, patients with type 2 myocardial infarction do not profit from invasive strategies, but rather from correction of the underlying oxygen supply and demand imbalance. In clinical practice, however, the differentiation between these two types is not always simple. ST-segment elevation and new severe regional wall motion abnormalities in a supply territory of a coronary artery are suspicious for type 1 infarction and call for a cardiological work-up. However, perioperative conditions, which present a danger of potentially fatal bleeding, must be considered before invasive strategies and antithrombotic therapies are started. Figure 3 proposes a treatment algorithm for patients suffering from perioperative myocardial injury.

## 6. Perioperative Cardiac Medication

In general, established cardiac medication should be continued perioperatively. This is particularly true in patients with heart failure. If additional cardiac drugs are necessary based on the cardiological work-up, they should be introduced weeks before surgery. Addition of cardiac medication immediately before surgery should be avoided.

Many patients with cardiac conditions take antiplatelet or antithrombotic drugs. Due to the risk of potentially fatal bleeding, the application of these drugs must be interrupted or changed in many cases. A comprehensive overview of this topic can be found in [41].

## 7. Conclusions

In patients at risk of cardiac issues undergoing non-cardiac surgery, robust evidence shows a strong association between elevated cardiac biomarkers (mainly cardiac troponin and the natriuretic peptides BNP and NTpro-BNP) and adverse outcomes. Incorporating biomarkers into the process of risk stratification and optimization offers an important opportunity to decrease perioperative morbidity and mortality. Despite its biological plausibility, prospective evaluation is warranted.

## Figures and Tables

**Figure 1 jcm-09-00351-f001:**
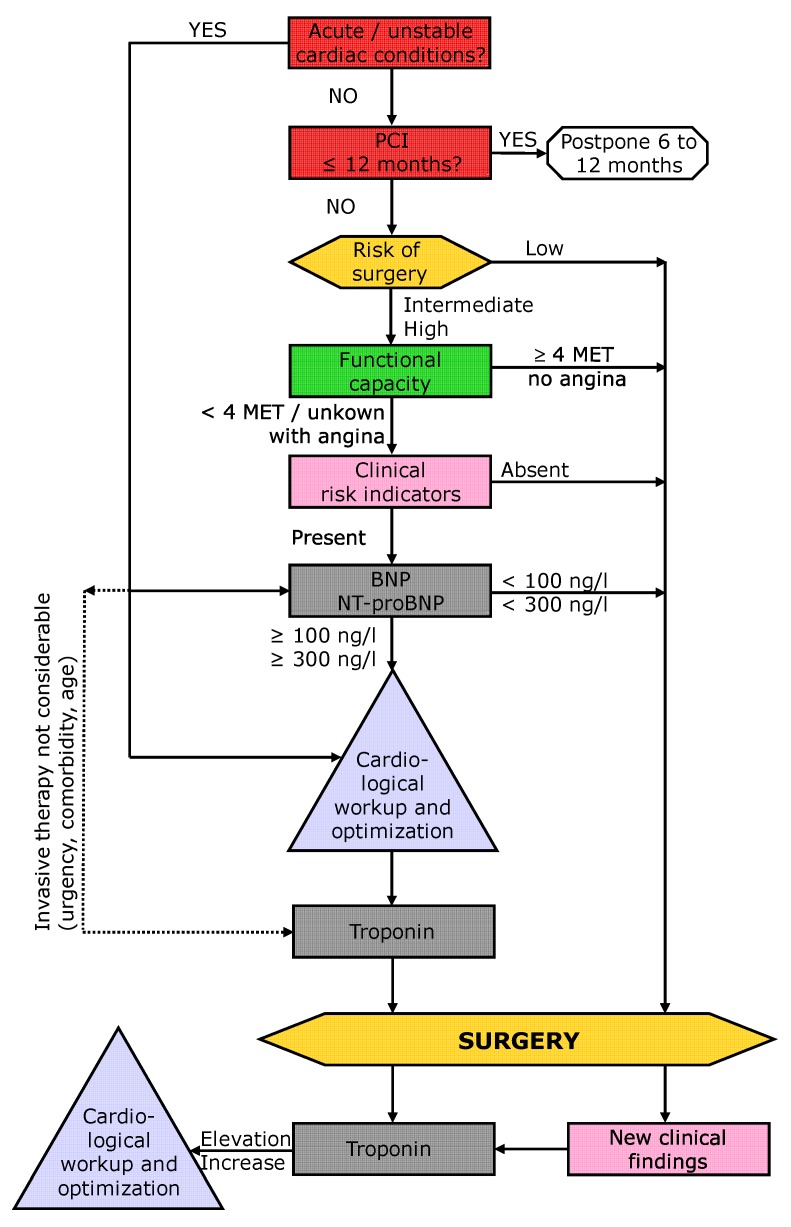
Swiss algorithm for risk stratification prior to elective non-cardiac surgery; modified from Filipovic M, Kindler C, and Walder B. Swiss Med Forum. 2018;18(5152):1078-80. https://doi.org/10.4414/smf.2018.03440. With the kind approval of the Swiss Medical Publishers EMH. PCI: Percutaneous Coronary Intervention. metabolic equivalent; BNP: brain natriuretic peptide.

**Figure 2 jcm-09-00351-f002:**
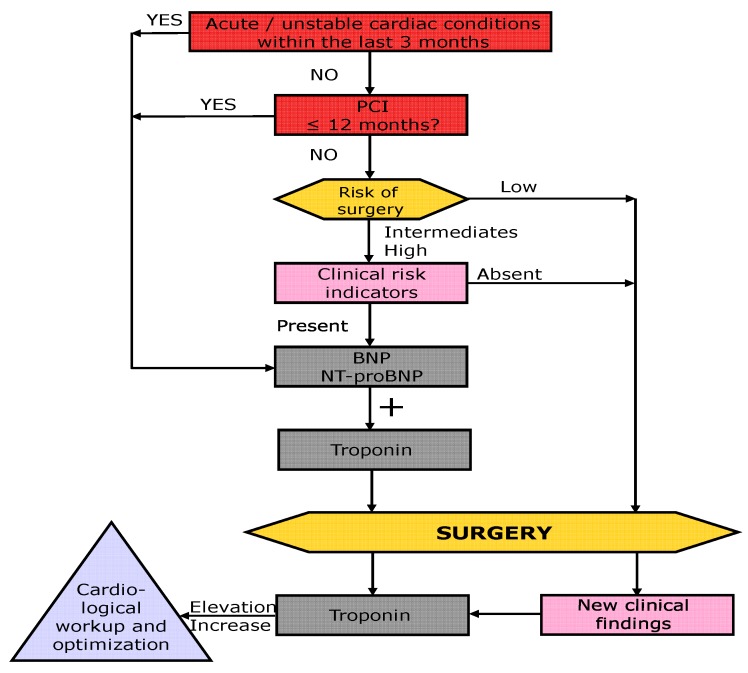
Swiss algorithm for risk stratification prior to emergent non-cardiac surgery; modified from Filipovic M, Kindler C, and Walder B. Swiss Med Forum. 2018;18(5152):1078-80. https://doi.org/10.4414/smf.2018.03440. With the kind approval of the Swiss Medical Publishers EMH. PCI: Percutaneous Coronary Intervention.

**Figure 3 jcm-09-00351-f003:**
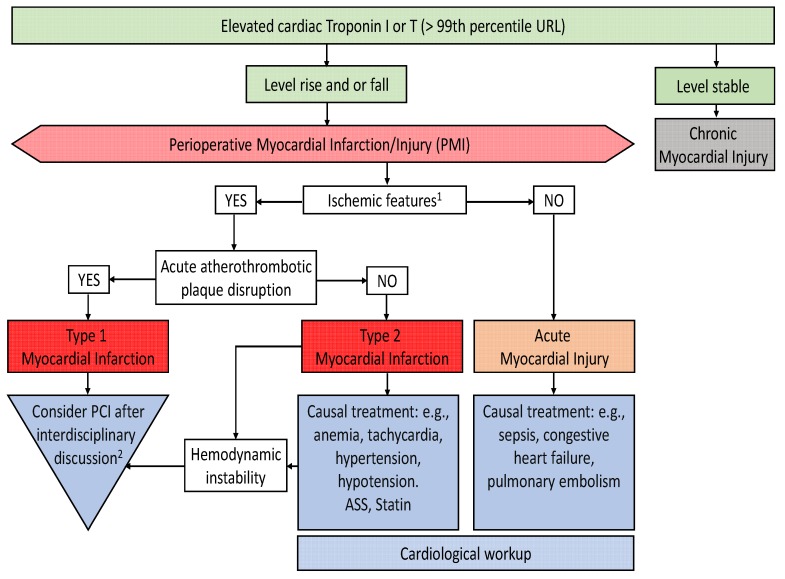
Proposed algorithm for perioperative myocardial injury/infarction based on the fourth definition of myocardial infarction [24]. ^1^ Ischemic features include signs or symptoms of acute myocardial ischemia, new ischemic ECG changes and/or development of pathological Q waves, imaging evidence of new wall motion abnormalities consistent with ischemic etiology. ^2^ Consideration of risk of bleeding after surgery. URL: Upper Range Limit; PCI: Percutaneous Coronary Intervention; ASS: acetylsalicylic acid.

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
