# Peer review of "Biomarker-Based Preoperative Risk Stratification for Patients Undergoing Non-Cardiac Surgery"

_jcm, 2020, doi:10.3390/jcm9020351_

Round 1

Reviewer 1 Report

The authors reviewed the literature on biomarker-based risk stratification for patients undergoing non-cardiac surgery. They provide an overview of current evidence and propose algorithms. The manuscript flows well and the flowcharts are simple to follow. I suggest the following to improve the manuscript: 

Methods section. It is unclear how the authors reviewed the current literature. What was omitted from this discussion? Were all relevant data sources evaluated? How was this done?  Discussion. The authors mention the Canadian Cardiovascular Society guidelines and how the biomarkers are placed in the center. However, this is not discussed further and the algorithm is not presented. Instead, the recently published Swiss guidelines are presented and leaves the reader wondering what "placed into the center" means and how that is different from the Swiss algorithm. I am also curious if the authors would recommend re-arranging the Swiss algorithm so the biomarker check is placed before functional measurement, especially given their language, starting at line 93.  Figure 3 - would the authors be able to add how often the troponin should be checked? After how many measurements is it considered rising/falling? Does that matter? That piece is unclear in that figure.

Author Response

Manuscript jcm-696950: “Biomarker-based preoperative risk stratification for patients undergoing non-cardiac surgery” for the upcoming special issue of the Journal of Clinical Medicine”

Dear Reviewer

Thank you for reviewing our above mentioned manuscript; we are grateful for your helpful comments and critiques. On the following pages, we have responded to them in detail and described how we have used them to improve our manuscript. We have highlighted the changes, including content and references, within the manuscript in red colour.

It is unclear how the authors reviewed the current literature. What was omitted from this discussion? Were all relevant data sources evaluated? How was this done?

We did not perform a systematic review on the association of natriuretic peptides with perioperative risk. However, since the publication of the meta-analysis by Rodseth and colleagues in the J Am Coll Cardiol 2011 (PMID: 21777751) and our own systematic review (Lurati Buse G, et al, Anest Analg 2011; PMID: 21372274) to our knowledge all published studies confirmed the prognostic significance of natriuretic peptides in this context. A very recent example is the large substudy of the VISION trial published a couple of days ago (Duceppe E, et al, Ann Intern Medicine, 2019 [Epub ahead of print]; PMID: 31869834). We added this information and the reference to the manuscript.

The authors mention the Canadian Cardiovascular Society guidelines and how the biomarkers are placed in the center. However, this is not discussed further and the algorithm is not presented. Instead, the recently published Swiss guidelines are presented and leaves the reader wondering what "placed into the center" means and how that is different from the Swiss algorithm.

We are grateful for this comment and added a short description of the Canadian Guidelines to the manuscript. An important difference between the two approaches is the Swiss algorithm asking for a cardiological workup in case of signs of an increased cardiac risk, whereas the Canadian one omits this step. This fact is mentioned in the manuscript (page 4, line 186-91)

I am also curious if the authors would recommend re-arranging the Swiss algorithm so the biomarker check is placed before functional measurement, especially given their language, starting at line 93.

The Swiss algorithm places the measurement of natriuretic peptides after the assessment of the functional capacity because the functional capacity is determined by interviewing the patient about his daily activities and not (as the reviewer might have assumed) by an objective measurement. Accordingly, the steps in the algorithm preceding BNP are nearly entirely based on patient's history and the risk of the scheduled surgery. Time and resource consuming cardiological assessment is only recommended if the easy to obtain information (including BNP or NTproBNP) indicate higher risk and room for optimisation. Accordingly, we prefer not to change our algorithm.

Figure 3 - would the authors be able to add how often the troponin should be checked? After how many measurements is it considered rising/falling? Does that matter? That piece is unclear in that figure.

We agree with the reviewer that information on the frequency and duration of troponin surveillance would be useful. We added this information to the manuscript (daily for two to three days). An elevated troponin is defined as any value above the 99th percentile upper reference limit of the test. The "Fourth definition of myocardial infarction" does not give a definitive definition of "rise and fall", however, a meaningful change in the value could be one of > 20%. We added this information and the corresponding references (PMID 23878152; PMID 30649367) to the manuscript.

Reviewer 2 Report

The study entitled „Biomarker-based preoperative risk stratification for patients undergoing non-cardiac surgery” is a review article considering a  very important issue in perioperative medicine.  Overall, the manuscript is well prepared. However, some changes might be considered.

Comments:

Traditional Risk Stratification. The authors mentioned each element known form the literature in 4 sections. Please consider adding to this part Table or Figure. It can be more interesting for readers. Consider mentioning the ongoing POISE  3 study in which  200 ng/L value for NT-pro BNP  is in inclusion criteria. Figures 1 and 2. There are similar. Leave the second in the manuscript. Please check the comas in the manuscript. They are where they shouldn’t and vice versa.

Author Response

Manuscript jcm-696950: “Biomarker-based preoperative risk stratification for patients undergoing non-cardiac surgery” for the upcoming special issue of the Journal of Clinical Medicine”

Dear Reviewer

Thank you for reviewing our above mentioned manuscript; we are grateful for your helpful comments and critiques. On the following pages, we have responded to them in detail and described how we have used them to improve our manuscript. We have highlighted the changes, including content and references, within the manuscript in red colour.

      1. Traditional Risk Stratification. The authors mentioned each element known from the literature in 4 sections. Please consider adding to this part Table or Figure.

We agree with the reviewer that the 4-step in the “Swiss Algorithm” needs to be highlighted, especially in Figure 1. We integrated an additional explanatory information in the main text pointing out the corresponding colours used in Figure 1. Because this information is given in the text and the existing figure 1, we would politely propose to waive an additional figure.

Consider mentioning the ongoing POISE  3 study in which  200 ng/L value for NT-pro BNP  is in inclusion criteria.

Thank you for this hint. We added the information to the manuscript.

Figures 1 and 2. They are similar. Leave the second in the manuscript.

We feel that the reviewer might have overlooked the difference between these two figures: Figure 1 describes the approach in elective cases and figure 2 in urgent procedures. Accordingly, we would propose to leave both figures in the manuscript.

Please check the commas in the manuscript. They are where they shouldn’t and vice versa.

We have checked and corrected the comas.

We hope that the manuscript now meets with your expectations and that the changes make it suitable for publication in The Journal of Clinical Medicine.

Sincerely yours,

Timur Yurttas, on behalf of all authors